# Personalized aerosolised bacteriophage treatment of a chronic lung infection due to multidrug-resistant *Pseudomonas aeruginosa*

Thilo Köhler [1,2] ✉, Alexandre Luscher[1,2], Léna Falconnet[1,2], Grégory Resch[3], Robert McBride[4], Quynh-Anh Mai[4], Juliette L. Simonin[5], Marc Chanson [5], Bohumil Maco [2], Raphaël Galiotto [2], Arnaud Riat[6], Natacha Civic[7], Mylène Docquier[7], Shawna McCallin[8], Benjamin Chan[9] & Christian van Delden[1,2]

Bacteriophage therapy has been suggested as an alternative or complementary strategy for the treatment of multidrug resistant (MDR) bacterial infections. Here, we report the favourable clinical evolution of a 41-year-old male patient with a Kartagener syndrome complicated by a life-threatening chronic MDR *Pseudomonas aeruginosa* infection, who is treated successfully with iterative aerosolized phage treatments specifically directed against the patient's isolate. We follow the longitudinal evolution of both phage and bacterial loads during and after phage administration in respiratory samples. Phage titres in consecutive sputum samples indicate in patient phage replication. Phenotypic analysis and whole genome sequencing of sequential bacterial isolates reveals a clonal, but phenotypically diverse population of hypermutator strains. The MDR phenotype in the collected isolates is multifactorial and mainly due to spontaneous chromosomal mutations. All isolates recovered after phage treatment remain phage susceptible. These results demonstrate that clinically significant improvement is achievable by personalised phage therapy even in the absence of complete eradication of *P. aeruginosa* lung colonization.

Bacteriophages are viruses that infect and lyse susceptible host bacteria; their clinical use predates that of antibiotics[1]. Phage therapy (PT), which has remained in use in countries of Eastern Europe and the former Soviet Union throughout the 20th century, is experiencing a resurgence of interest worldwide due to the emergence and spread of multidrug (MDR) and pan-drug resistant bacterial strains[2]. The efficacy of PT has been documented in several case reports showing promising results both during topical and systemic applications. Phages have been used to treat surgical[3,4] and respiratory tract infections with MDR bacteria (reviewed in refs. 5–7), as well as recurrent bone and joint

[1]Service of Infectious Diseases, Geneva University Hospitals, Geneva, Switzerland. [2]Department of Microbiology and Molecular Medicine, University of Geneva, Geneva, Switzerland. [3]Center for Research and Innovation in Clinical Pharmaceutical Sciences (CRISP), Lausanne University Hospital (CHUV), Lausanne, Switzerland. [4]Felix Biotechnology, San Francisco, CA, USA. [5]Department of Cell Physiology and Metabolism, University of Geneva, Geneva, Switzerland. [6]Diagnostic Bacteriology Laboratory, Geneva University Hospitals, Geneva, Switzerland. [7]iGE3 Genomics Platform, University of Geneva, Geneva, Switzerland. [8]Department of Neuro-Urology Balgrist Hospital, Zurich, Switzerland. [9]Yale University, New Haven, CT, USA. ✉ e-mail: thilo.kohler@unige.ch

infections[8,9], and an aortic graft infection due to MDR *Pseudomonas aeruginosa*[10].

Only a few placebo-controlled randomized clinical trials (RCT) using PT have been published[11]. One study tempted intravesical phage cocktail application for urinary tract infections[12], while the PhagoBurn RCT compared standard-of-care sulfadiazine treatment to topical application of a phage cocktail for the treatment of *P. aeruginosa* burn-wound infections[13]. Both studies showed successful treatment in a subset of patients infected with phage-susceptible *P. aeruginosa* isolates.

*P. aeruginosa* remains a major cause of progressive lung function deterioration in cystic fibrosis patients, as well as in patients with chronic obstructive pulmonary disease or chronically infected bronchiectasis. Due to repeated or continuous administration of antibiotic therapies, the emergence of MDR isolates is frequently observed, leaving only limited conventional treatment options[8].

Here, we report an expanded-access treatment of a patient with a chronic lung infection using a single lytic bacteriophage selected for its activity against the patient's MDR *P. aeruginosa* isolate in a personalized-medicine approach. The application of phage via nebulization significantly improved the patient's condition and reduced the bacterial load in tandem with phage replication, as demonstrated directly in sputum samples. Our report further highlights the importance of initial bacterial screening at the population level for the design of a successful patient-tailored PT protocol.

## Results

### Patient history

The patient was a 41-year-old male with Kartagener syndrome. In 2014, the patient suffered a traumatic C7-D1 fracture leaving him tetraplegic. Chronic respiratory tract colonization by *P. aeruginosa* became apparent in 2018 and was complicated by monthly episodes of acute respiratory exacerbations requiring prolonged hospital stays for broad-spectrum intravenous (IV) antibiotic treatment. The resistance phenotype progressively worsened with MDR *P. aeruginosa* strains being isolated by July 2019. These recurrent exacerbations led to an almost continuous hospitalization starting in July 2019 (Fig. 1a). Computed tomography (CT) showed severe bilateral lower lung consolidations (D-126, Fig. 1b).

### Aerosolized phage treatment

Given the absence of significant improvement over a seven-month hospitalization and dependence on daily IV suppressive antibiotics, individualized phage therapy was attempted in March 2020 in addition to continuous IV antibiotic treatment. Phage vFB297, a member of the genus *Pakpunavirus* (Supplementary Fig. S1), showed lytic activity on the patient's latest MDR *P. aeruginosa* isolate D-100 (Fig. 1a and Supplementary Fig. S2). The phage was resuspended in 5 ml of PBS and administered through a nebulizer over 20 min. During five consecutive days, the patient received a daily dose of $5 \times 10^9$ PFU of phage vFB297,

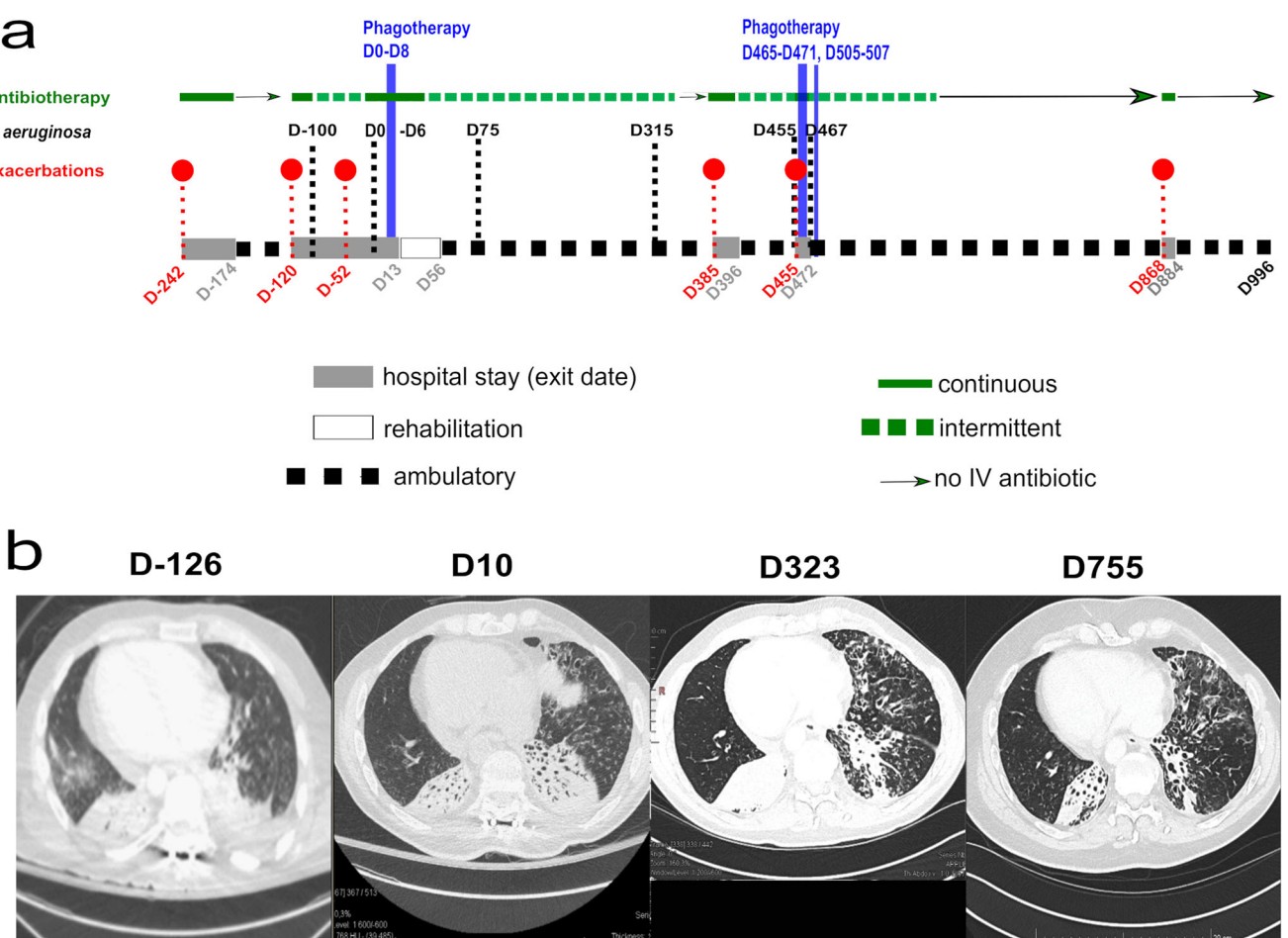

**Fig. 1 | Patient treatment, follow-up and sample collection. a** Antibiotic treatments are indicated by green horizontal lines and phage therapies by blue vertical lines. *P. aeruginosa* isolates selected for genotypic and phenotypic analysis are shown above the black-dotted vertical lines. Numbering of isolates and events is in reference to the day of the first phage treatment onset (D0). Episodes of exacerbations are indicated by red dots. Hospital and rehabilitation stays are indicated on the horizontal bar. **b** Computed tomography (CT) lung scans were performed at the days indicated, with respect to phage treatment onset (D0).

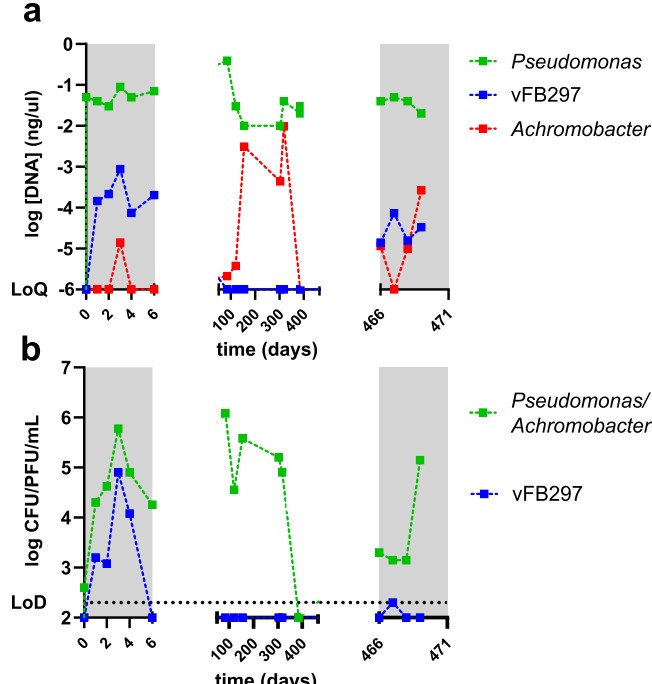

**Fig. 2 | Bacterial and phage load in sputum samples during and between phage courses.** Samples are numbered with respect to the first phage administration (D0). The first sputum sample was taken 15 min before the first phage dose, while all other samples were obtained 7–8 h after phage administration. **a** Sputum samples were analyzed for the presence of *P. aeruginosa*, *A. xylosoxidans* as well as phage DNA by qPCR on total genomic DNA. **b** In parallel, combined viable counts of *P. aeruginosa* and *A. xylosoxidans* (CFU) were determined on cetrimide agar plates in triplicate, which do not allow distinguishing between these bacterial species. Infective phage particles (PFU) were counted on phage indicator plates in triplicate. LoQ limit of quantification, LoD limit of detection. Phage treatment courses are indicated by gray-shaded rectangles.

followed two days later by two additional doses. An initial sputum sample was obtained 15 min before the first phage administration (here defined as day 0, D0, Fig. 1a), while subsequent respiratory samples were recovered 7–8 h after each phage administration. During the first two days of phage therapy, the patient's oxygen saturation dropped, concomitant with an initial rise in C-reactive protein (CRP) and a single febrile event. All subsequent phage treatments were well-tolerated, and no further side effects were observed. Globally, the patient's clinical condition improved rapidly. On day 5 post-treatment initiation, the patient was free from respiratory obstruction, and consequently no respiratory sample could be collected. A CT-scan performed after ten days showed stable bilateral lower lung condensations (D10, Fig. 1b). Given his improvement, the patient was transferred to a rehabilitation center (D13) on continuous antibiotic treatment, downgraded to intermittent, thrice weekly suppressive IV therapy (D24) and was finally discharged home (D56).

**Clinical course following phage therapy**
As an outpatient, the patient further gained autonomy and was free from respiratory exacerbations. From the sputum, various *P. aeruginosa* isolates with MDR phenotypes, as well as an *Achromobacter xylosoxidans*, were regularly cultivated. The initial plan to provide him with aerosolized phages for 5 days every month in a preventive approach could not be pursued because of the subsequent waves of the COVID-19 pandemic. After 10 months, a control chest CT showed progressive clearance of the left lower lung infiltrates with the persistence of the right-sided consolidation (D 323, Fig. 1b). Discontinuation of the intermittent systemic antibiotic therapy was attempted.

Unfortunately, he relapsed (D385) and required a short course of continuous antibiotics. Following a second exacerbation, he received a 5-day course of daily phage vFB297 aerosol administration (D465-471), two administered as inpatient and three as outpatient. Again, the phage treatment was well-tolerated and the clinical benefit was rapid with less dyspnea and sputum production. Three further phage doses were administered outpatient (D505-507, Fig. 1a). In January 2022, a new attempt to stop systemic antibiotics was made. A control chest CT-scan showed a significant reduction of the right-sided consolidation and almost cleared left-sided infiltrate (D 755, Fig. 1b). Following a brief respiratory relapse requiring daily IV antibiotics (D868), antibiotic therapy could be stopped in August 2022 and the patient remained free of antibiotics and respiratory exacerbation since then until the last follow-up in December 2022 (D 996, Fig. 1a). He resumed his professional activity during summer 2022.

**Inpatient microbiological evolution**
Genomic DNA isolated from sputum samples was used to detect the bacterial and phage loads during and after the two-phage courses. Plating of sputum samples on cetrimide agar and spotting of filtered supernatants on indicator plates allowed the determination of CFU and PFU counts, respectively (Fig. 2). Genomic *Pseudomonas* DNA content was similar during (Fig. 2a, left and right panels) and between phage courses (Fig. 2a, center panel). The diagnostic bacteriology laboratory also detected transient colonization by *A. xylosoxidans*, which was confirmed by qPCR on genomic DNA isolated from sputum samples (Fig. 2a). During the first treatment course, phage DNA was detected 24 h after phage administration (D1) and its concentration increased until day 3 (D3) (Fig. 2a, left panel). Analysis of the same sputum samples by CFU and PFU counts, reflecting viable bacteria and infective phage particles, respectively, revealed a dynamic picture consistent with inpatient phage replication during the first treatment course. Indeed, both *P. aeruginosa* CFU and phage vFB297 PFU increased in the sputum until D3 followed by a 2- and 3-log decrease at D6, i.e at the end of the first treatment course (Fig. 2b, left panel), respectively. Neither phage DNA nor PFUs could be detected between the two-phage treatments (Fig. 2a, b, center panel, respectively). During the second treatment course, phage DNA (Fig. 2a, right panel) as well as PFUs (Fig. 2b, right panel) were detected in sputum, albeit at lower levels compared to the first treatment course.

**Phenotypic heterogeneity of *P. aeruginosa* isolates from sputum samples**
Phage vFB297 (Fig. 3a) was initially selected on isolate D-100, collected 100 days prior to the first phage therapy course. Surprisingly, isolates collected in the sputum sample taken immediately before the first phage dose administration showed susceptible (D0.2, D0.3, D0.4), intermediate (D0.6), and phage-resistant (D0.1, D0.5) phenotypes (Fig. 3b). However, all isolates tested were able to produce infective vFB297 phage particles in liquid medium, resulting in a 1 to 5-log increase in phage titers (Supplementary Fig. S3). Subsequent isolates obtained during the first phage treatment period and thereafter were all susceptible to phage vFB297 (Fig. 3c and Supplementary Fig. S3). Minimum inhibitory concentration (MIC) determinations performed on at least three isolates per sputum sample revealed heterogenous susceptibility profiles for beta-lactam antibiotics (aztreonam, ceftazidime, meropenem) with susceptible and resistant isolates present in the same sputum sample (Fig. 3d–f and Supplementary Table S1). After the first phage treatment course, only phage-susceptible but beta-lactam-resistant clones were isolated, thus suggesting a progressive enrichment of this bacterial subpopulation.

**Genome analysis of patient *P. aeruginosa* isolates**
Based on distinct phage and antibiotic susceptibilities (Supplementary Table S1), seven isolates (D-100, D0.1, D6.1, D75.1, D315.1,

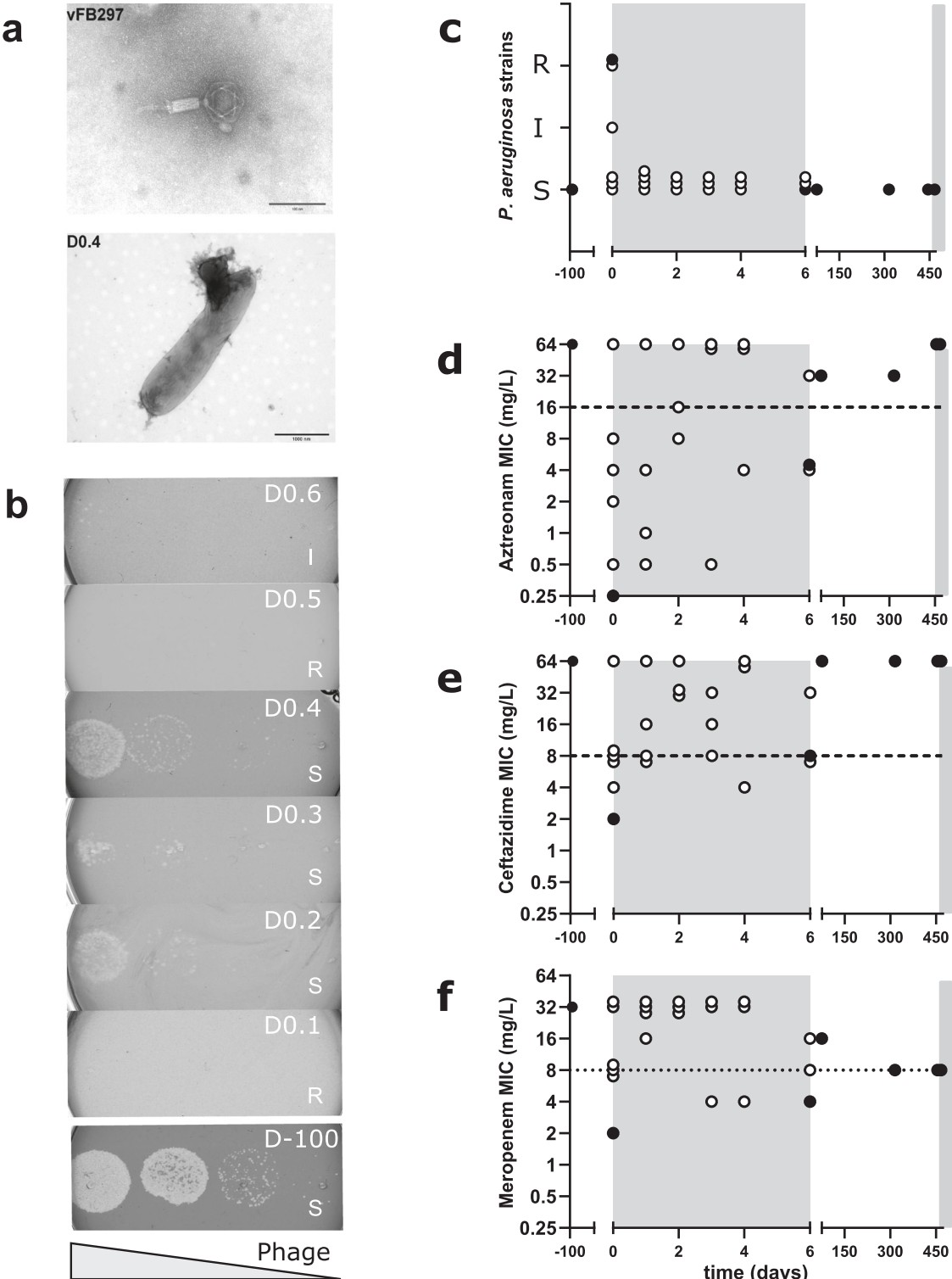

**Fig. 3 | Susceptibility of *P. aeruginosa* isolates to phage vFB297 and beta-lactam antibiotics.** Transmission electron microscopy images of (**a**) *Pakpunavirus* vFB297 (top panel) and susceptible D0.4 bacterial cell lysed by vFB297 phage particles (bottom panel). **b**, **c** Susceptibility of patient isolates to phage vFB297 was determined by plaque assays (for full dataset, see Supplementary Fig. S2). S susceptible (plaques visible in phage dilutions), I intermediate (plaques visible only in undiluted phage), R resistant (no plaques visible). **d**–**f** Susceptibilities to beta-lactam antibiotics of *P. aeruginosa* isolates from consecutive sputum samples. MIC determinations were performed on three occasions, yielding similar results. MIC breakpoints (EUCAST 2019) for beta-lactams are shown by dashed lines. Each symbol represents a single *P. aeruginosa* isolate. The seven isolates selected for WGS are represented by closed symbols. Phage treatment courses are indicated by gray-shaded rectangles.

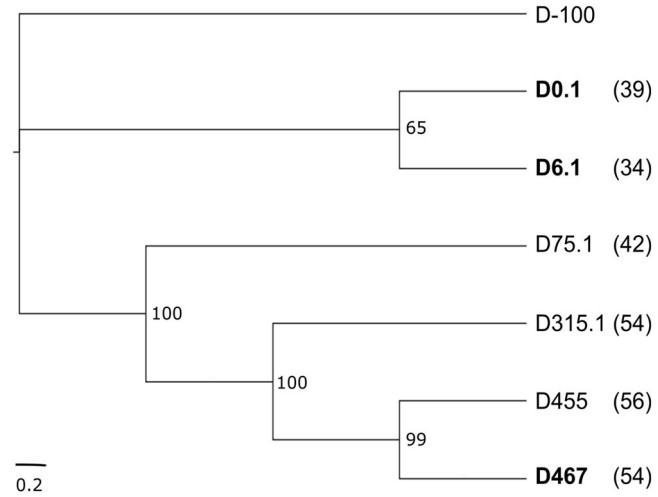

**Fig. 4 | Phylogeny of patient isolates.** All isolates sequenced belong to the same DLST-type (15-29). Isolates D0.1, D6.1, and D467 (shown in bold) were isolated during the first and second phage treatment, respectively. Phylogeny results are based on SNP analysis using vcf2phylip v2.6 and FigTree v1.4.4. Numbers in parentheses indicate the number of additional cumulative SNPs compared to the D-100 isolate as determined by WGS (Supplementary Dataset 1). Bootstrap values are shown at the branching points.

D455, D467) depicted in Fig. 1 were selected for genomic and phenotypic characterization. Double locus sequence typing (DLST)[14] revealed the same DLST-type (15-29) in all isolates, suggesting colonization by a single genotype before, during, and after phage therapy (Table 1).

The isolates were submitted for whole-genome sequencing (WGS) and the genome of isolate D-100 was de novo assembled yielding a predicted genome size of 6,545,241 bp that was comparable to the *P. aeruginosa* reference strain PA14 (6,537,648 bp). Alignment of the genome of isolate D-100 with PA14 revealed 53,139 SNPs (Supplementary Dataset 1) as well as genomic deletions ranging in size from 0.25 kbp to 61.7 kbp (Supplementary Dataset 2). All sequenced isolates, including the initial D-100 strain, harbored a nonsense mutation in *mutS* (codon W708) and two amino acid substitutions each in MutT and in MutL, previously described to be responsible for a hypermutator phenotype in isolates from CF patients[15,16]. Among the mutations present only in the beta-lactam resistant isolates (D-100, D75.1, D315.1, D455, and D467), WGS analysis identified a frameshift (fs) mutation in the *mpl* gene encoding an UDP-N-acetylmuramate-L-alanyl-gamma-D-glutamyl-meso-diaminopimelate ligase involved in recycling of peptidoglycan in *P. aeruginosa*[17]. Mutations in *mpl* have been associated with high expression of the chromosomal AmpC cephalosporinase in both laboratory and clinical isolates[17,18]. This agrees with higher *ampC* expression levels in the beta-lactam-resistant isolates as determined by qRT-PCR (Table 1). All sequenced genomes displayed a frameshift in *ampD* (83delA) as well as several amino acid substitutions in AmpR and in AmpC compared to PA14. Furthermore, a frameshift mutation in the carbapenem-specific porin gene, *oprD*, (348delC) likely explains the decreased susceptibility of the isolates to meropenem in comparison to PAO1 (Table 1 and Supplementary Dataset 1).

The phylogenetic analysis deduced from the WGS showed two clades, both derived from the common D-100 ancestor isolate; the first formed by the early isolates (D0.1 and D6.1) and the second by the later isolates (D75.1 to D467) (Fig. 4). SNP analysis seems to indicate a continuous accumulation of mutations in the sequential patient isolates (Fig. 4).

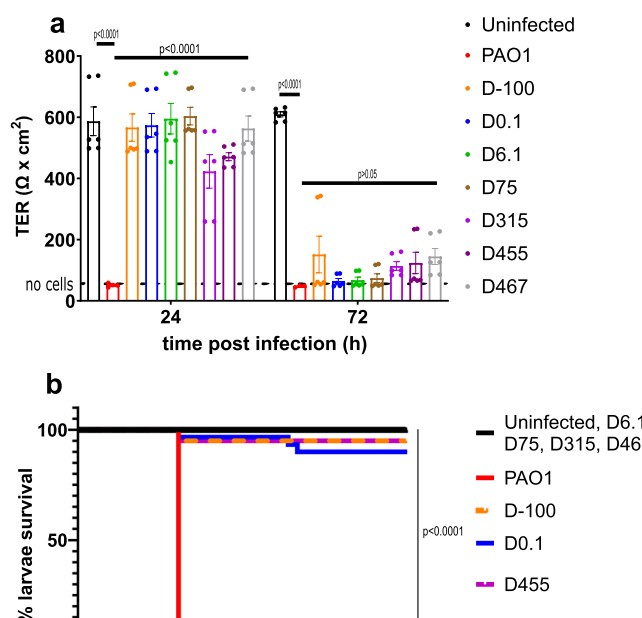

**Fig. 5 | Cytotoxicity and virulence of *P. aeruginosa* isolates. a** Calu-3 cells were infected with PAO1 or patient isolates and trans-epithelial resistance was determined at 24 h p.i. and 72 h p.i. Values represent mean and standard error of the mean (SEM) of three independently performed experiments in technical duplicates. **b** PAO1 and patient isolates were tested for their virulence in a *Galleria mellonella* infection model. The Kaplan–Meier plots represent the survival of 20–30 larvae that were used for each strain. Larvae survival was followed up to 72 h. PAO1 was the virulent reference strain (>90% killing at 72 h p.i.). The significance of differences of TER (**a**) or larvae survival (**b**) were assessed using one-way ANOVA with Dunnett multi-comparison test (**a**) or log-rank test (**b**), by comparison to the positive control strain PAO1.

## Phages do not select for *P. aeruginosa* variants with increased virulence

To assess potential phenotypic changes during phage therapy courses, we further characterized isolates with respect to virulence traits, including rhamnolipid production, motility, and biofilm formation capacity (Table 1). Compared to the initial isolate D-100, all later isolates showed increased biofilm formation capacity. None of them produced rhamnolipids. While all isolates were swimming proficient, only isolates D-100, D0.4, D6.1, and D75.1 showed a weak swarming phenotype. This trait was lost in later isolates (D315.1, D455, D467) and is thus unlikely to result from phage therapy. Later isolates showed improved growth compared to early isolates in LB medium (Supplementary Fig. S4).

Cytotoxicity of the *P. aeruginosa* isolates was assessed on Calu-3 respiratory airway epithelial cells cultivated on Transwell filters[19] by measuring the trans-epithelial electrical resistance (TER). The cytotoxic PAO1 control strain decreased TER 24 h post infection (p.i.) (from 600 to -50 $\Omega \times cm^2$), indicating disruption of epithelial layer integrity (Fig. 5a, red bar). The patient isolates were non-cytotoxic at 24 h and produced values either identical or slightly lower than the uninfected control well (Fig. 5a, black bar). However, at 72 h p.i., all patient isolates decreased TER values almost to the level of PAO1, suggesting a delay in the expression of their cytotoxic capacity (Fig. 5a).

To further assess the virulence of consecutive isolates, we used the *Galleria mellonella* infection model. Although this wax moth lacks an adaptive immune response, its innate immune system is comparable with that of vertebrates, making it a suitable model to study

**Table 1 | Genotypic and phenotypic characterization of *P. aeruginosa* patient isolates**

| Strain/isolate[a] | Genotype | | | Phenotype | | | | |
|---|---|---|---|---|---|---|---|---|
| | DLST | *ampC* | *mpl* | *ampC* (fold exp) | β-Lactam suscept.[b] | Biofilm (%) | Swimming | Swarming |
| PAO1 | 16-4 | wt | wt | 1 | S | 100 | ++ | ++ |
| D-100 | 15-29 | wt | **M38fs** | 3079 ± 5850 | **R** | 124 ± 60 | + | + |
| D0.1 | 15-29 | wt | wt | 5.2 ± 1.8 | S | 509 ± 163 | + | − |
| D0.2 | 15-29 | ND | ND | ND | S | 307 ± 214 | + | − |
| D0.3 | 15-29 | ND | ND | ND | S | 159 ± 67 | + | − |
| D0.4 | 15-29 | ND | ND | ND | S | 132 ± 21 | + | + |
| D0.5 | 15-29 | ND | ND | ND | S | 338 ± 115 | + | − |
| D0.6 | 15-29 | ND | ND | ND | **R** | 304 ± 92 | + | − |
| D6.1 | 15-29 | wt | wt | 0.54 ± 0.30 | S | 164 ± 121 | + | + |
| D75.1 | 15-29 | wt | **M38fs** | ND | **R** | 247 ± 127 | + | + |
| D315.1 | 15-29 | V239A | **M38fs** | ND | **R** | 252 ± 240 | + | − |
| D455 | 15-29 | V239A, V356I | **M38fs** | 17.9 ± 9.7 | **R** | 495 ± 289 | + | − |
| D467 | 15-29 | V239A, V356I | **M38fs** | 42.3 ± 29.3 | **R** | 377 ± 159 | + | − |

[a]Isolate nomenclature designates the day of sampling (D), starting with the first day of phage administration (D0), and is followed by the number of the individual isolate, when more than one colony was recovered from the sputum sample (e.g., D0.1, D0.2, etc). D-100 designates the isolate used for phage screening, recovered 100 days before the phage therapy onset (D0).
[b]Beta-lactam-resistant isolates (R) and mutations in Mpl are highlighted in bold.

virulence in acute infections[20]. Virulence was defined as increased host mortality resulting from *P. aeruginosa* infection. While the PAO1 reference strain killed 90% of larvae within 24 h, all patient isolates tested, whether recovered before or after phage therapy, resulted in 90% survival of infected larvae (Fig. 5b). These results agreed with the Calu-3 cell culture assay, except that at 72 h, no additional larvae were killed. We conclude that the virulence phenotype of the patient's isolates did not evolve over time and was not influenced by the phage treatment.

## Discussion

Here we report a personalized phage treatment of a patient chronically infected by an MDR *P. aeruginosa* strain. Only few studies have reported the administration of phages by inhalation to treat *P. aeruginosa* infections[21–24]. In these studies, phages were additionally administered IV in combination with antibiotic therapies. Although infections could be resolved in all cases, eradication of *P. aeruginosa* was achieved only in a ventilator-associated acute pneumonia case[21], but not in patients with chronic infections[22,23]. Even without the eradication of the pathogen, our patient, presenting a life-threatening clinical situation due to a chronic MDR *P. aeruginosa* infection, showed a drastic improvement of his general condition after receiving a patient-tailored phage therapy. Notably, this improvement went in parallel with an objective progressive clearance of lung consolidations visualized on sequential chest CT-scans, which allowed progressive transition from daily to intermittent systemic antibiotics and finally a complete stop of systemic antimicrobial therapies. Except for a transient oxygen desaturation during the first phage therapy course, no adverse events related to the phage treatment were observed, allowing subsequent outpatient phage administration. The measured endotoxin concentrations were above the threshold levels indicated for IV applications (5 EU/kg/h) and could explain the brief inflammatory response and fever episode after the first phage administration. However, the patient's chronic *P. aeruginosa* colonization and hence continuous exposure to *P. aeruginosa* LPS could explain that endotoxins present in the phage preparation were well-tolerated when administered by nebulization.

Antibiotics alone could not sufficiently reduce the burden of the MDR *P. aeruginosa* in a severely altered lung environment characterized by bronchiectasis and reduced sputum removal due to the deficient ciliary clearance, combined with a severe respiratory muscular dysfunction linked to his spine lesion. We hypothesize that a phage-susceptible *P. aeruginosa* population, predominating in the lower respiratory tract and the bronchiectasis, was responsible for the chronic inflammatory response and the frequent infectious exacerbations. By reducing the burden of this population, the phage/antibiotic combination ultimately reduced chronic inflammation and prevented infectious exacerbations. In addition, the combined treatment may have also helped to kill slow-replicating, biofilm-trapped bacteria that are normally impossible to eradicate with antibiotics alone, as highlighted recently by in vitro eradication of non-replicating PAO1 cells with combined meropenem and phage treatment[25]. However, we cannot exclude the existence of a non-accessible phage-resistant population in the lower airways, which might explain the failure to eradicate *P. aeruginosa* colonization. Indeed, obtaining clinical samples from the lower respiratory tract via bronchoalveolar lavage remains an invasive procedure that requires clear clinical indications, which were not fulfilled here. Of note, the combined subjective clinical improvement with the objective continuing clearance of lung consolidations, reduction of respiratory exacerbations, and eventual discontinuation of systemic antibiotics makes a placebo effect unlikely.

To the best of our knowledge, our inpatient data are the first to demonstrate in vivo phage replication in a patient's lung. After an initial increase in phage titers during the first three days after phage administration, a drop in viable phage counts occurred (Fig. 2b) and was concomitant with a decrease in the bacterial population. The discrepancy between phage loads determined in the lung by qPCR and live counts (blue lines in Fig. 2a vs b) might be due to phage DNA released from lysed bacterial cells or phage particles and detectable only by qPCR. We also cannot exclude some variability in content of the sputum samples that might have affected quantification and sample comparison.

The isolate available for initial phage screening (D-100) was an already host-adapted clone that probably colonized the patient for several months. WGS revealed that this isolate, as well as all tested subsequent isolates, were hypermutators, harboring multiple mutations and genomic deletions compared to the reference strain PA14. Hypermutators are typically found in 30% of CF patients and contribute to rapid initial adaptation to the particular CF lung environment[16]. The lung environment of our patient is comparable to a CF lung environment with missing ciliated movements and accumulation of thick mucus. Hypermutators not only drive antibiotic resistance but also favor the emergence of beneficial compensatory

mutations in frequently targeted genes like *ampC* and *ftsI*[26]. Indeed, all isolates even before phage treatment showed multiple distinct mutations in *ampC* and *ampR*. However, broad-spectrum beta-lactam resistance, including meropenem, clearly correlated with *ampC* overexpression due to a mutation in Mpl, a protein involved in peptidoglycan recycling (Table 1)[18,26]. The combined antibiotic/phage therapy seemed to enrich the pre-existing MDR population (Fig. 3d–f). However, we did not observe emergence of phage-resistant isolates in the respiratory samples recovered during or after the two-phage treatments (Fig. 3c).

The dominance of hypermutators likely explains the phenotypic heterogeneity with respect to phage and antibiotic susceptibility, observed at the onset of phage therapy (D0). This pinpoints the complexity in selecting the active phage or designing phage cocktails when faced with an infection due to a heterogenous bacterial population coupled with the difficulty of obtaining fully representative clinical samples. Phenotypic analysis showed no evolution towards more virulent variants during or after phage therapy. All tested isolates including the initial D-100 isolate revealed low virulence phenotypes both in the epithelial cell culture and the *G. mellonella* models.

Overall, our study showed the safety and efficacy of aerosolized personalized bacteriophage treatment with concomitant antibiotics for a chronic MDR *P. aeruginosa* life-threatening lung infection. Although a complete microbiological eradication was not achieved, the patient's condition improved significantly, allowing hospital discharge and discontinuation of systemic antibiotic treatments. Our study, although limited to a single observation, clearly shows the importance of screening multiple *P. aeruginosa* isolates in chronically infected patients and supports the use of specifically tailored phage therapy.

## Methods

### Ethical statement
According to the CARE guidelines and in compliance with the Declaration of Helsinki principles, the compassionate phage therapy was authorized by the medical direction of the Geneva University Hospitals, and the patient provided written informed consent for the phage protocol therapy and publication of his case as a scientific manuscript. All methods and experimental protocols were carried out in accordance with the approved guidelines.

### Sample and isolate collection
All samples and isolates were recovered from patient sputum, obtained after bronchial physiotherapy, when required. Sputum samples were the only samples clinically indicated, consequently no other clinical samples (i.e., nasal swabs, bronchoalveolar lavage fluid) were collected. Sputum samples were split on site. One aliquot was sent to the local bacteriology diagnostic laboratory of the University Hospitals Geneva (HUG) and the remainder was transported on ice to the nearby research laboratory for bacterial and phage counting and sample storage. An aliquot of the sputum sample was spread on cetrimide agar plates for single colony isolation and incubated at 37 °C. *P. aeruginosa* isolates were sequentially collected during 467 days. Bacterial isolates were identified by MALDI-TOF. Isolates are named according to their day of isolation, with reference to the first day of phage administration (day 0, D0). Isolates were stored in 20% (v/v) glycerol at −80 °C.

### Phage preparation
Strain PAO1 was grown in TSB medium overnight. 50 ml of TSB medium were inoculated with 500 µl of the overnight culture and incubated (250 rpm, 37 °C) until early mid-exponential phase. In total, 10 µl of phage vFB297 were added and the culture was incubated overnight at 37 °C. The culture was transferred into a 50 ml Falcon tube and centrifuged at 1717 × *g* for 20 min at 4 °C. The supernatant was filter sterilized (0.22 µm). 10 ml of the phage preparation was added to an

Amicon Ultra 100 K column (Sigma-Aldrich) and centrifuged at 1717 × *g* for 30 min at 4 °C. The concentrated phage solution was recovered from the column and titered on a lawn of PAO1 bacteria. Endotoxins were determined by the Limulus amoebocyte lysate assay by a third-party laboratory and were below 1250 EU/ml. Phage was prepared in 5 ml of sterile saline solution at a titer of $10^9$ PFU/ml (first treatment) and $10^8$ PFU/ml (2nd treatment). The sterility of the phage preparations was verified by spreading 100-µl aliquots on LB and blood agar plates and incubation at 37 °C for 24 h.

### Sample treatment for DNA extraction and determinations of bacterial and phage counts
A 1 ml aliquot of the sputum sample (usually 2–4 ml) was removed with a positive displacement pipette (Gilson Microman® E) and added to 0.4 ml of 87% (v/v) glycerol, and frozen immediately at −80 °C (protected). Another 1 ml aliquot was sampled similarly and frozen directly at −80 °C (native). A tenfold volume of 0.9% NaCl was added to the remaining sputum sample, which was homogenized by pipetting up and down using a 10-ml pipette. In all, 5 ml of this suspension was centrifuged in a 15-ml Falcon tube at 966 × *g* for 5 min. The supernatant was transferred into a 10-ml syringe and filter sterilized through a 0.22-µm filter to obtain at least 1 ml of filtrate, which was kept at 4 °C. From the resuspended pellet (5–10 ml), tenfold serial dilutions were prepared in 0.9% NaCl and spread on cetrimide agar plates and incubated at 37 °C for at least 24 h to determine *P. aeruginosa* and *A. xylosoxidans* CFUs. The susceptibility of isolates to phage was determined by spotting tenfold dilutions of phage vFB297 on a lawn of host bacteria using the double agar overlay plaque assay (1% agar in the bottom and 0.4% agar in the top layer prepared in LB medium).

### Phage replication determination
Phage vFB297 was added at an MOI of 0.02 to *P. aeruginosa* isolates in 1 mL LB medium, and incubated at 37 °C. At each time point, 200 µl of each mixture was centrifuged for 5 min at 16,200×*g*. Overall, 150 µl of supernatant was transferred to another Eppendorf tube and 8 µl of 98% chloroform was added to each tube to kill the bacteria. In total, 100 µl of this sterilized solution was serially diluted and spotted on a lawn of PAO1 for PFU counts after overnight incubation at 37 °C.

### Genomic DNA extraction and quantitation
Total genomic DNA (gDNA) from homogenized sputum samples (200 µl) was benzonase-treated (SIGMA) and extracted (Qiagen DNeasy kit) according to the manufacturer's guidelines. qPCR was performed with a Quantinova SYBR green kit (QIAGEN) on 20 ng of total gDNA with the following conditions: one cycle of 95 °C for 5 min, then 40 cycles of 95 °C for 5 s and 60 °C for 10 s. Primer pairs shown in parenthesis were used at 1 µM to detect gDNA of *P. aeruginosa* (rpsL-F/R), Phage vFB297 (SP-Phi30.1/2) and *A. xylosoxidans* (AX-F1/R1) (Supplementary Table S2).

### Genotyping of *P. aeruginosa* isolates
Isolates were genotyped by double locus sequence typing (DLST)[14]. Primers ms172.forward and ms172.reverse, as well as ms217.forward and ms217.reverse were used for the amplification of the highly variable genomic loci ms271 (400 bp) and ms217 (350 bp) of *P. aeruginosa* (Supplementary Table S2). The DNA sequences of the obtained PCR products were mapped on the DLST database (www.dlst.org), yielding a double two-digit genomic code.

### Whole-genome sequencing
DNA from seven *P. aeruginosa* isolates was extracted using the Qiagen tissue extraction kit according to the manufacturer's protocol. Whole-genome sequencing was performed using the Nextera protocol on an Illumina MiSeq machine at the Genomic Core Facility of the University of Geneva. The Illumina MiSeq (total reads: 9,987,633) paired-end

300 bp reads were adapter and quality trimmed (mean PHRED quality score >39) with Sickle v.1.3. The average coverage (with PCR duplicates marked) was >73. Contigs were aligned on the reference genome of PA14 (NC_008463.1), which showed maximal coverage for the majority of contigs. Reads for isolate D-100 were de novo assembled into contigs with SPAdes v.3.13.1 and evaluated without reference with QUAST v.5.0.2. The antimicrobial resistance genes were detected with ResFinder 4.1.

### RNA extraction and cDNA preparation

For RNA extraction, strains were grown in LB medium until reaching an $OD_{600}$ of 1. A 500 µl culture aliquot was added to 1 ml of RNA Protect solution (Qiagen), and RNA was extracted with RNeasy kit according to the manufacturer's protocol (Qiagen). Genomic DNA was removed by treatment with RNase-free DNase (Promega). cDNA was then obtained by reverse transcription of 500 ng of total RNA with Improm-II reverse transcriptase (Promega) according to the manufacturer's instructions.

### Quantitative real-time PCR

Gene expression was assessed on two independent cDNA synthetized from two independent RNA preparations. Expression of *ampC* was quantified using Quantinova SYBR Green PCR kit (Qiagen), with primers specific for *ampC* and the *rpsL* housekeeping gene as normalization control (Supplementary Table S2). qPCR was performed on 20 ng of cDNA in a Rotor-Gene 3000 (Corbett Research) under the following conditions: 5 min at 95 °C, 5 s at 95 °C, and 10 s at 60 °C (40 cycles). A melt-curve analysis was performed (60–99 °C) to confirm the presence of a single amplification product.

### Minimal inhibitory concentration (MIC) assays

MIC determinations were performed according to CLSI guidelines in Mueller–Hinton broth medium[27].

### Microtiter plate biofilm assay

After overnight culture in LB medium (37 °C), bacteria were grown in M63 medium supplemented with 0.05% casamino acids, 0.4% arginine, 1 µg/mL vitamin B12, 1 µg/mL vitamin B1, 1 mM $MgSO_4$ (20 µl overnight culture in 180 µl of supplemented M63 medium) in a 96-well plate. The plate was incubated for 24 h at 37 °C in static conditions and the absorbance was measured at 600 nm. The attached cells (biofilms) were stained with crystal violet[28]. The optical density of cultures was measured at 550 nm in a plate reader (BioTek Synergy H1). All experiments were performed in biological duplicates using PAO1 as the reference strain.

### Motility assays

Swimming motility was measured as follows: a colony from a fresh LB plate, was stab inoculated into a 0.3% LB agar plate and incubated at 37 °C for 18 h. Swarming motility was determined by inoculating a single colony from a fresh LB plate, onto a 0.5% agar plate medium (1 × M8 medium supplemented with 0.2% glucose and 0.05% glutamate as nitrogen source) and incubated for 24 h at 37 °C[29].

### Rhamnolipid production

Five µl of overnight culture in LB medium were spotted on a rhamnolipid agar medium plate composed of M8 salts 1×, 0.2% glucose, 2 mM $MgSO_4$, 0.0005% methylene blue, 0.1% glutamate, 0.02% cetyltrimethylammonium bromide and solidified with agar (1,6% final concentration)[29,30]. Plates were incubated overnight at 37 °C, followed by 24 h at room temperature and 3–5 h at 4 °C. Rhamnolipid production is indicated by a blue-violet halo forming around the colony. The diameter of this halo, corresponding to a precipitate of rhamnolipids, was measured in mm and compared to the one obtained with PAO1.

### Calu-3 airway epithelial cell infections

Cytotoxicity was assessed by infecting a monolayer of Calu-3 cells (ATCC HTB55, Manassas, VA, USA) with *P. aeruginosa* as described[19]. This cell line is not part of misidentified cell lines. Briefly, overnight LB-grown bacteria were resuspended in saline buffer (NaCl 0.9%, HEPES 10 mM, $CaCl_2$ 1.2 mM) and adjusted by serial dilution in the saline buffer to approximately $10^5$ CFU/ml. Ten µl of this dilution (approx. inoculum of $10^3 \pm 10^1$ CFUs, MOI of 0.002) were added apically to the transwell filter containing $5 \times 10^5$ Calu-3 cells. Ten microliters of saline buffer were added to uninfected control cells, while strain PAO1 was used as a reference. Filters were incubated at 37 °C with 5% $CO_2$ for 24 h and 72 h[19]. At this point, bacterial growth was determined by adding apically 200 µl of saline buffer to the transwell filter. A 100 µl aliquot was removed apically and diluted tenfold to perform plate counts on LB agar plates. Trans-epithelial electric resistance (TER) was measured as before[19] using chopstick electrodes and a volto-meter (EVOM, World Precision Instruments, Inc.).

### *Galleria mellonella* in vivo infection model

Infection of wax moth larvae was performed as described[31,32], with the following modifications. Batches of wax moth larvae were purchased from Expressbait (Basel, Switzerland). Larvae weighing $250 \pm 50$ mg were kept in the dark at 37 °C for 24 h before infection, for temperature acclimatization. Batches of ten larvae were used on two separate occasions for each isolate. Overnight bacterial cultures were centrifuged, cell pellets were washed and diluted 30,000× in PBS to reach an inoculum size of $10^{5 \pm 0.5}$ CFU/ml. Ten microliters of bacterial suspension in PBS ($10^{3 \pm 0.5}$ CFU total) were injected in the last left proleg using a 0.5 ml insulin syringe (Omnican). Larvae survival was monitored for 72 h. Larvae were considered dead when melanized and/or not responding to poking with a tip. Survival data were pooled by conditions and represented as Kaplan–Meier plots. Comparison between PBS and infected conditions were tested for significance using the log-rank test (GraphPad Prism).

### Transmission electron microscopy

Phage vFB297 and *P. aeruginosa* bacteria were mixed at a MOI of 30 in 0.9% NaCl and incubated for 30 min at RT. For negative staining, a 5 µl sample was applied to a freshly glow-discharged (for 1 min) copper 200-mesh electron microscopy grid coated with a 1% formvar plastic support film and stabilized with an additional thin carbon film. After 1 min adsorption, the grid was blotted with filter paper and washed three times on droplets of MilliQ water. Grids were stained for 1 min with a 1% aqueous uranyl acetate solution and finally washed with MilliQ water for 10 s and air-dried. Electron micrographs were collected with a Tecnai 20 TEM (FEI, Netherland) operated at 80 kV acceleration voltage and equipped with a side-mounted CCD camera (MegaView III, Olympus Imaging Systems) controlled by iTEM software (Olympus Imaging Systems) at the Electron Microscopy Facility (PFMU), Medical Centre, Faculty of Medicine, University of Geneva, Switzerland.

### Statistics

Statistical analyses were performed and graphs were generated using the GraphPad Prism software (v. 9.5.1). Images were treated with ImageJ (v1.53).

### Reporting summary

Further information on research design is available in the Nature Portfolio Reporting Summary linked to this article.

## Data availability

The raw data generated in this study have been deposited to Zenodo: https://doi.org/10.5281/zenodo.8005315 (https://zenodo.org/search?

page=1&size=20&q=8005315). The phage sequence has been submitted to GenBank under the accession number #OQ921398. Genomic sequencing reads of the patient isolates are available at the Sequence Read Archive (SRA) BioProject PRJNA980116 (http://www.ncbi.nlm.nih.gov/bioproject/8980116). Source data are provided with this paper.

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

## Acknowledgements

We acknowledge the invaluable help from physicians and the nursing staff of the HUG for sample collection, as well as Georgia Mitropoulou, especially in the context of COVID-19. We are grateful to the Queen Astrid Hospital (Bruxelles, Belgium) for phage screening. We thank Mylène Docquier and Natacha Civic from the iGE3 genomic platform from the Medical Faculty of the University of Geneva for performing WGS and sequence analyses and for helpful discussions. We express our gratitude to Angela Huttner for careful review of the manuscript. This work received partial financial support from the Swiss National Science Foundation grant 32473B_179289 to C.v.D.

## Author contributions

T.K., C.v.D., and A.L. designed and supervised the work, collected samples, analyzed the data, and wrote the manuscript. A.L., L.F., T.K., J.L.S., Q.M., R.G., and B.M. performed all experimental work. M.C. and J.L.S

provided and cultured cell lines and analyzed data. R.M., B.C., and G.R. performed phage screenings and R.M., Q.M., and B.C. prepared phage. A.R. collected and analyzed bacterial isolates. N.C. and M.D. performed WGS and bioinformatic analyses. S.M. co-directed the phage working group at the HUG and was instrumental for obtaining phage vFB297.

## Competing interests

R.M. is co-founder and CEO of Felix Biotechnology, San Francisco, USA. The remaining authors declare no competing interests.
