## [Peer Review File · Nature Communications]

REVIEWER COMMENTS

Reviewer #1 (Remarks to the Author):

The manuscript by Köhler et al. describes a single case report of a phage therapy intervention for treatment of a persistent MDR *P. aeruginosa* infection. The persistent nature of the infection and multiple rounds of antibiotic therapy prior to phage treatment supports the idea that the phage treatment played some role in clinical improvement. Strengths of the work include collection and sequencing of multiple bacterial samples, which shows the infection was a single acquisition of a pathogen followed by diversification by mutation. Interestingly, phage resistance did not appear during treatment and bacterial virulence and antibiotic sensitivity did not change during or after treatment.

COMMENTS

1. The phage vFB297 was used for treatment but its characteristics are not described in the manuscript. The phage is described as a member of the Pakpunavirus genus, so I suspect that the phage genome has already been sequenced. At a minimum, the phage genome sequence should be deposited in an INSDC repository (NCBI or EMBL) and some basic features, including relationships to other known phages, described in the manuscript.
2. Somewhere in the manuscript (e.g., the heading for Table 1) should describe the rationale for the strain naming scheme so they can be confidently interpreted by the reader. It looks like the scheme is [day#].[isolate#], this should be clarified.
3. Fig. 1 has a number of issues. A magenta star in the legend denotes “Rehabilitation” but this symbol does not appear on the figure. The “no antibiotic” arrow feature does not also need the word “STOP” above it, as its meaning is already conveyed in the legend. The CT images should be enlarged significantly; they could be a separate panel or made large enough so that they are as wide as the timeline image (i.e., each CT image is $\sim \frac{1}{4}$ the width of the whole image). If possible, the green antibiotic treatment line should be disaggregated into the different drugs used for treatment, rather than all lumped together into a single “Antibiotherapy” line.
4. In Fig. 2, it is not clear if points at the X-axis detection limit are counted or if this means that no CFU or PFU were observed (i.e., counts were below the detection limit). There should be an X-axis title for both figures.

5. The electron micrographs presented in Fig. 3a do not add much to the manuscript and could be removed. The TEM of the phage shows a contracted myophage that is of low quality, and the disrupted cell does not tell us anything about the phage or host other than that the phage can lyse it, which is already known.
6. For Fig. 3C, state the criteria for determining sensitive/intermediate/resistant for the phages.
7. In Fig. 3F, there are two lines for breakpoints, it is not clear what this means.
8. Fig. 4 should have a scale bar and node bootstrap values.
9. In Fig. 5, panels A and B could be merged into one figure with side-by-side bars for 24 and 72 h; color coding for each isolate is not needed as the axes are labeled.
10. In the Methods, primers are named but I could not identify a table containing a list of primer sequences. This needs to be included with the manuscript as a main or supplementary table.
11. Throughout the manuscript the mu (μ) symbol is not displayed correctly.

Reviewer #2 (Remarks to the Author):

The paper gives a nice exposition of a simple clinical case, in which a colonised/infected gentleman with background CF suffered for months with *Pseudomonas* superinfection including classic hypermutator phenotypes, as expected in this setting and well demonstrated here.

He was offered phage therapy in desperation, with good clinical response, and then again later, again with good response.

The clinical story is strongly suggestive that phage was successful but provides no definitive evidence. I think the clinical anecdote is too long and could be greatly simplified.

Of note is the evidence of intra-pulmonary/ intra-airways phage amplification after inhalation only, and the fact that recovering *Pseudomonas* populations are not significantly affected in terms of phage or antibiotic adaptation. These data address an important assumption in phage therapy – that is, that the

nature of the interaction is so mature that efficacy can be expected without eradication and that relapse can be expected without resistance, so that retreatment is entirely feasible and practicable.

The emphasis in the abstract is about right I think. The case report element, currently around a thousand words, could be abbreviated in the text, perhaps by leaning more on Fig 1 and its legend, as n=1 phage therapy anecdotes have little value other than to illustrate the phage-bacterial interplay, which is the more important element of the story for this readership.

The evidence presented of phage amplification in the airways is important but does not appear to be accompanied by evidence of phage in blood. That does not appear to have been tested and is probably worth a mention because the question of whether Pseud DNA spills into blood or phages are detectable in light of the hint at an inflammatory response early on.

The early clinical decline suggestive of an inflammatory response to phage therapy is not discussed but a Jarisch-Herxheimer type reaction vs simply too much endotoxin needs to be considered in the text. His calculated endotoxin per dose is 5mL over 20 mins @ (less than) 1250EU/mL means that he may have received ~6000EU over 20 mins, which is for a 70kg man means ~85 EU/kg over 20 mins. This is well in excess of the pyrogenic threshold of 5EU/kg/hr which we are advised (e.g. by FDA) not to exceed for intravenous preparations and (logically) for nebulised preparations. It is noteworthy that this was so well tolerated thereafter. This should be briefly discussed.

In Fig 2 the conflation of the ox pos cat pos Pseud and Achromobacter is understandable but makes interpret v difficult and this should just be explained if it cant be corrected.

The virulence studies are well presented but probably could be more summary as well. The most important data are the wax moth and cell culture data. The assertion of pyoverdine synthesis as an explanation for slow growth could be reference by alluding to any one of many refs that show the relationship between pyoverdine synthesis and bacterial killing.

TEM images add colour rather than data and could also be trimmed away to save space, at the editors discretion.

Reviewer #3 (Remarks to the Author):

[1] Was there any possibility to try to identify the poly-microbial species present?

[2] *Pseudomonas aeruginosa* is cleared by 300-400 days, but then comes back by around 466 onwards. Where is coming from? Is it an internal source, a niche that harbours a smaller number of undetected bacteria or is the source external? Previous studies have shown that the nasopharynx may act as a potential reservoir for re-infecting the lower airways. Did the authors look at upper respiratory tract colonisation?

[3] I don't see the point of comparing the clinical isolates to PA01 in figure-5. PA01 will be very different to these host adapted strains, so the value of this comparison is minimal. Why not compare the clinical isolates from this patient to other isolates from other patients at different stages of host adaptation?

[4] The Galleria models does not really tell us anything new or useful about these strains. The model itself is an acute infection model so it is not surprising that PA01 kills the larvae while biofilm forming host adapted clinical strains do not. I think the ideal model to investigate these clinical strains would have been a mouse model of chronic lung infection, one which allows persistence of infection, biofilm formation and interactions with both humoral and cellular immunity.

Reviewer #4 (Remarks to the Author):

The authors report in details on a single case of phage use in a patient with an MDR pseudomonas infection. Although similar reports have been published before, it is refreshing to see such details in a case report.

My overall suggestions/comments:

1. The manuscript is extremely long for a case report, esp as a type of 'communication'. Some of the details may be more suited as a story, but may lack scientific value. I would recommend:

(a) Reduction in number of, or details of previous studies on phage therapy: no need to review all the previous experience in a communication as the readers would be lost long before reading the actual report.

(b) Many details are not important: e.g. p127-132: specific dates are not important. If authors want to make a point on duration, they can change dates to number of days, which may be more useful in giving context to the 'story'. Or line 108-113: Can simply indicated phage therapy was initiated after consent was obtained from patient using phage vFB297... Words like 'last resort' makes the reading more entertaining but also distracting.

(c) Authors should summarize the clinical course more succinctly, focusing not on providing as much detail as possible, but thinking about what is most important in their goal of the report.

(d) Please comment on the phage safety: one of the main issues are quality control of phage and potential cytokines etc in the preparation

(e) Be careful in claiming 'quantitative' comparison in sputum samples: notorious that there are huge variability in content in different sputum samples (vs BAL, for instance)

There are quite a few minor points that are not too important but can make the reader keep pausing and affect the readability an otherwise enjoyable journey of a patient's story. A few examples:

(a) Line 87: no need to say 'congenital': this is understood

(b) Line 88: sinus inversus is not a 'complication' of Kartagener but really part of the syndrome

(c) No need to report the doses of antibiotics

Responses to the reviewer's comments are shown in blue and newly added or modified text is highlighted by quotation marks.

REVIEWER COMMENTS

Reviewer #1 (Remarks to the Author):

The manuscript by Köhler et al. describes a single case report of a phage therapy intervention for treatment of a persistent MDR *P. aeruginosa* infection. The persistent nature of the infection and multiple rounds of antibiotic therapy prior to phage treatment supports the idea that the phage treatment played some role in clinical improvement. Strengths of the work include collection and sequencing of multiple bacterial samples, which shows the infection was a single acquisition of a pathogen followed by diversification by mutation. Interestingly, phage resistance did not appear during treatment and bacterial virulence and antibiotic sensitivity did not change during or after treatment.

COMMENTS

1. The phage vFB297 was used for treatment but its characteristics are not described in the manuscript. The phage is described as a member of the Pakpunavirus genus, so I suspect that the phage genome has already been sequenced. **At a minimum, the phage genome sequence should be deposited in an INSDC repository (NCBI or EMBL)** and some basic features, including relationships to other known phages, described in the manuscript.

Response: We thank the reviewer for this comment. Indeed, the phage had been sequenced (phage genome size of 90,516 bp) We have submitted the genome sequence to DFAST (<https://dfast.ddbj.nig.ac.jp/>) for annotation. As with many phage genomes, most of the ORFs are annotated as hypothetical proteins. A complete map with generated annotations has been added to supplementary data as a novel Figure (Figure S1). The relationship to homologous phages has been added to the legend of Figure S1 to avoid interrupting the flow in the main manuscript. The phage genome, including annotations, has been deposited to NCBI GenBank and is available as accession # OQ921398.

We have added the following figure legend, which contains all information requested by the reviewer:

“Fig. S1. Genomic map of phage vFB297. BLAST search in the NCBI database retrieved *Pseudomonas* phages 908-1, SRT6, PaYy-2 and vB_PaeM_SCUT-S2 as showing more than 94% nucleotide identity with vFB297. These phages are annotated as belonging to the genus Pakpunavirus. Their complete genome sizes (90 – 95 kbp) are comparable to those of phage vFB297 (90,516 bp). The genome sequence of vFB297 was annotated using DFAST (https://dfast.ddbj.nig.ac.jp/help_annotation) and the genome map generated with SnapGene. Arrows in green identify genes with functional predictions, while all other arrows indicate genes encoding hypothetical proteins. The phage genome was submitted to NCBI GenBank (accession # OQ921398). “

2. Somewhere in the manuscript (e.g., the heading for Table 1) should describe the rationale for the strain naming scheme so they can be confidently interpreted by the reader. It looks like the scheme is [day#].[isolate#], this should be clarified.

Response: The footnote to Table 1 in the submitted version already described the designation and meaning of isolate names, but this might not have been visible enough:

“Isolate nomenclature designates day of sampling (D), starting with first day of phage administration (D0), and is followed by the number of the individual isolate, when more than one colony was recovered from the sputum sample (e.g. D0.1, D0.2, etc). D-100 designates isolate for phage screening recovered 100 days before the first phage administration.”

To make this naming more visible to the reader we have added the following sentence to the legend of figure 1, where isolate names appear for the first time:

“Bacterial isolates as well as sampling days and CT scan dates are numbered in reference to the day of first phage treatment onset (D0).”

We have also added to the Methods section (first paragraph) the sentence:

“Isolates are named according to their day of isolation starting with the first day of the initial phage administration (day 0, D0).”

3. Fig. 1 has a number of issues. A magenta star in the legend denotes “Rehabilitation” but this symbol does not appear on the figure. The “no antibiotic” arrow feature does not also need the word “STOP” above it, as its meaning is already conveyed in the legend. The CT images should be enlarged significantly; they could be a separate panel or made large enough so that they are as wide as the timeline image (i.e., each CT image is ~ ¼ the width of the whole image). If possible, the green antibiotic treatment line should be disaggregated into the different drugs used for treatment, rather than all lumped together into a single “Antibiotherapy” line.

Response: We thank the reviewer for this comment. To improve the reading of figure 1, we have modified it as follows:

- Stop was removed
- “rehabilitation period” indicated by a star was replaced by a neutral white rectangle and indicated as such in the figure legend
- CT scans were enlarged and fitted below the timeline as suggested by this reviewer
- However, we decided to keep the green “generic” antibiotic treatment line. Given that the patient received several antibiotics and with different treatment periods, we felt it would be too confusing for the reader to detail all treatment regimens; if requested the different antibiotics could be added to the supplementary data

4. In Fig. 2, it is not clear if points at the X-axis detection limit are counted or if this means that no CFU or PFU were observed (i.e., counts were below the detection limit). There should be an X-axis title for both figures.

Response: We agree with the reviewer that this was confusing. For the qPCR assay in panel 2A, we used the term limit of quantification (LoQ), meaning that our qPCR assay detected a signal but the values were not in a range for allowing precise quantification. For the CFU and PFU determinations in panel 2B, our limit of detection (LoD) is 200 CFU or PFU/ml. We indicated this by a stippled line at 2×10^2 , meaning that in samples below this line, we did not count any CFU or PFU. We added definitions for abbreviations “LoQ” and “LoD” to the revised figure legend and also added the x-axis title (time (days)) to both panels.

5. The electron micrographs presented in Fig. 3a do not add much to the manuscript and could be removed. The TEM of the phage shows a contracted myophage that is of low quality, and the disrupted cell does not tell us anything about the phage or host other than that the phage can lyse it, which is already known.

Response: We partially agree with the reviewer; indeed, such phage images have been shown previously. However, we would prefer to keep the images of the phage as an illustration and leave the decision to remove them at the editor’s discretion.

6. For Fig. 3C, state the criteria for determining sensitive/intermediate/resistant for the phages.

Response: In the submitted version we have given the following definition for phage susceptibility in the legend to Fig. 3:

Susceptibility of patient isolates to phage vFB297 was determined by plaque assays (full data set see Fig. S2); S, susceptible (plaques in phage dilutions); I, intermediate (plaques only in undiluted phage); R, resistant (no plaques).

We feel that these “arbitrary” criteria are sufficiently clearly defined and further supported visually by the supplementary Figure S2. We have only added the word “visible” behind the word “plaques”, hoping this helps further clarifying the definition.

7. In Fig. 3F, there are two lines for breakpoints, it is not clear what this means.

Response: We thank the reviewer for this comment. Indeed, these are based on the former EUCAST classification (before 2020), which still included the intermediate (I) resistance category. Therefore, the upper line delineates resistant (R) from intermediate and the bottom

line delineated intermediate (I) from susceptible (S) populations. To be consistent with the other figure panels, we removed the bottom line and therefore distinguish only between resistant (R) and (I)/(S) isolates.

8. Fig. 4 should have a scale bar and node bootstrap values.

Response: To improve figure 4 as suggested we have prepared a new version of the phylogeny tree of the isolates (Fig. 4) using the most recent converter vcf2phylip v.2.6 and FigTree v1.4.4 generating the bootstrap values, which indicate high confidence levels (> 60%) for the branching points. The new figure 4 also shows the scale bar as requested; the figure legend was modified accordingly.

9. In Fig. 5, panels A and B could be merged into one figure with side-by-side bars for 24 and 72 h; color coding for each isolate is not needed as the axes are labeled.

Response: We agree with the reviewer's comment that for space and simplicity reasons, the initial two panels A and B should be merged into a single graph (now panel A). This also allows removing the isolate name labelling on the x-axis. We also now show the individual data points for each replicate. Panel B showing data from the *Galleria mellonella* remains unchanged.

10. In the Methods, primers are named but I could not identify a table containing a list of primer sequences. This needs to be included with the manuscript as a main or supplementary table.

Response: We apologize for this missing information. We had prepared such a Table, but apparently omitted to join it during submission. The new Table (Table S4) showing the primer sequences used is now part of the supplementary information.

11. Throughout the manuscript the mu (μ) symbol is not displayed correctly.

Response: We have checked the μ symbol throughout the manuscript and made replacements where necessary.

Reviewer #2 (Remarks to the Author):

The paper gives a nice exposition of a simple clinical case, in which a colonised/infected gentleman with background CF suffered for months with *Pseudomonas* superinfection including classic hypermutator phenotypes, as expected in this setting and well demonstrated here.

He was offered phage therapy in desperation, with good clinical response, and then again later, again with good response.

The clinical story is strongly suggestive that phage was successful but provides no definitive evidence. I think the clinical anecdote is too long and could be greatly simplified.

Of note is the evidence of intra-pulmonary/ intra-airways phage amplification after inhalation only, and the fact that recovering *Pseudomonas* populations are not significantly affected in terms of phage or antibiotic adaptation. These data address an important assumption in phage therapy – that is, that the nature of the interaction is so mature that efficacy can be expected without eradication and that relapse can be expected without resistance, so that retreatment is entirely feasible and practicable.

Response: we thank the reviewer for his positive global comment.

Comments:

1. The emphasis in the abstract is about right I think. The case report element, currently around a thousand words, could be abbreviated in the text, perhaps by leaning more on Fig 1 and its legend, as n=1 phage therapy anecdote have little value other than to illustrate the phage-bacterial interplay, which is the more important element of the story for this readership.

Response: We agree with this comment. The initial case description was meant to provide the reviewers with a complete clinical picture, but this made the description indeed too long. As suggested, we have significantly shortened the patient's history and treatment description and removed "anecdotal" descriptions, thereby reducing the length of this section from 991 words to 597 words in the revised version.

2. The evidence presented of phage amplification in the airways is important but does not appear to be accompanied by evidence of phage in blood. That does not appear to have been tested and is probably worth a mention because the question of whether Pseud DNA spills into blood or phages are detectable in light of the hint at an inflammatory response early on.

Response: We thank the reviewer for this comment. Indeed, we have not performed blood sampling and can therefore not assess whether phage was present in the blood. At the time of phage therapy, the patient was clinically stable, although “antibiotic-dependent”, and without signs of respiratory exacerbation (acute pneumonia) or bacteraemia. If during the phage therapy, *P. aeruginosa* or other lung colonizing bacteria would have gained access to the bloodstream, the clinical symptoms and systemic inflammatory markers (CRP) would have been much stronger. We therefore hypothesise that the rise in CRP and short initial fever resulted from a local intra-bronchial inflammatory response either to endotoxin provided with the phage preparation or to the lysis of *Pseudomonas* cells secondary to the phage therapy.

3. The early clinical decline suggestive of an inflammatory response to phage therapy is not discussed but a Jarisch-Herxheimer type reaction vs simply too much endotoxin needs to be considered in the text. His calculated endotoxin per dose is 5mL over 20 mins @ (less than) 1250EU/mL means that he may have received ~6000EU over 20 mins, which is for a 70kg man means ~85 EU/kg over 20 mins. This is well in excess of the pyrogenic threshold of 5EU/kg/hr which we are advised (e.g. by FDA) not to exceed for intravenous preparations and (logically) for nebulised preparations. It is noteworthy that this was so well tolerated thereafter. This should be briefly discussed.

Response: This an interesting point that we had not discussed in detail. We cannot exclude a Jarisch-Herxheimer reaction but since the patient was continuously treated with antibiotics and had been colonised in his lungs for more than 4 years by *P. aeruginosa*, we estimate that the patient had been exposed continuously to lysed bacteria and hence to *P. aeruginosa* LPS. The amount of contaminating LPS originating from the *P. aeruginosa* strain used for phage production is therefore likely negligible in the context of aerosolized administration. This would of course be different for IV phage applications. Our observation suggests that much higher endotoxin levels might be tolerated upon lung nebulization compared to IV phage treatments. We have added the following sentence to the discussion section:

“The measured endotoxin concentrations were above the threshold levels indicated for IV applications (5 EU/kg/h) and could explain the brief inflammatory response and fever episode after the first phage administration. However, the patient’s chronic *P. aeruginosa* colonization and hence continuous exposure to *P. aeruginosa* LPS could explain that endotoxins present in the phage preparation were well-tolerated when administered by nebulization. “

4. In Fig 2 the conflation of the ox pos cat pos Pseud and Achromobacter is understandable but makes interp v difficult and this should just be explained if it can’t be corrected.

Response: We agree with the reviewer, we have not tempted to differentiate the two organisms based on their close metabolic requirements and biochemical similarity. Accordingly, we have added the following wording to the figure legend of Fig. 2:

“..., which does not allow distinguishing between these bacterial species.”

5. The virulence studies are well presented but probably could be more summary as well. The most important data are the wax moth and cell culture data. The assertion of pyoverdine

synthesis as an explanation for slow growth could be referenced by alluding to any one of many refs that show the relationship between pyoverdine synthesis and bacterial killing.

Response: As suggested by this reviewer as well as reviewer 1 (item 9), we have combined panel A and B of Fig. 5. Panel A summarizes the cell culture data and panel B the Galleria model results. We have also removed the two panels of pyoverdine production in supplementary Fig. S4 and stated in the results section:

“None of the isolates tested produced pyoverdine (data not shown), the main *P. aeruginosa* siderophore, which agrees with the large genomic deletions identified by WGS in the pyoverdine synthesis operons (Table S3) and could explain their slow growth *in vitro*”.

6. TEM images add colour rather than data and could also be trimmed away to save space, at the editor’s discretion.

Response: As discussed in our response to comment 5 of reviewer 1: we would like to keep the images as an illustration for those readers who are less familiar with bacteriophages and their size proportion compared to their bacterial host as shown on the second image below. However, we leave the final decision to the editor’s discretion.

Reviewer #3 (Remarks to the Author):

[1] Was there any possibility to try to identify the poly-microbial species present?

Response: This is an interesting point raised by the reviewer. However, we consider that this requires a novel set of experiments and bioinformatic analyses, which we consider should be part of a follow up study and are beyond the scope of this case report.

[2] *Pseudomonas aeruginosa* is cleared by 300-400 days, but then comes back by around 466 onwards. Where is it coming from? Is it an internal source, a niche that harbours a smaller number of undetected bacteria or is the source external? Previous studies have shown that the nasopharynx may act as a potential reservoir for re-infecting the lower airways. Did the authors look at upper respiratory tract colonisation?

Response: We thank the reviewer for mentioning this point. We rechecked our data and found that during figure preparation two data points (D303 and D315) in panel B corresponding to combined CFUs (green dots) were not correctly represented. This is now corrected in the new Figure 2B. The data also agree with the semi-quantitative estimates made by the clinical diagnostic laboratory. Both show a decrease at D395-396 in the load of *Pseudomonas/Achromobacter*, so this does not change the interpretation of the data and the reviewer’s comment remains valid.

We have not performed any oral, nasal or sinus swabs for this patient. Indeed, for CF-patients, presence of *P. aeruginosa* mainly in the sinuses has been suggested as a potential source for recolonization of the CF-lung after lung transplantation. We would expect that the genetic defect causing decreased mucociliary clearance associated with the Kartagener syndrome might also apply to the sinus epithelium and could favour bacterial colonization. However, we do not have experimental evidence for this. From DLST typing we can say that the later strains

have the same genotype (Table 1) and differ only by a few SNPs compared to the isolates collected before D315. This is also supported by the phylogeny of the isolates shown in Fig. 4. Hence an environmental acquisition of novel isolates seems unlikely.

[3] I don't see the point of comparing the clinical isolates to PAO1 in figure-5. PAO1 will be very different to these host adapted strains, so the value of this comparison is minimal. Why not compare the clinical isolates from this patient to other isolates from other patients at different stages of host adaptation?

Response: This point is well taken. However, we needed the PAO1 strain as a validation for our virulence assays. PAO1 works reliably in our hands both for the Calu-3 airway epithelial cell assays and the Galleria model. It allows standardising the results and comparisons between different isolates from a single patient. Most of the CF-isolates we have tested in these models, also those from other CF-patients, showed decreased virulence. Hence it is impossible to use such isolates for the validation of the assays. We therefore prefer to keep the data (Fig. 5B) as shown in the initially submitted version.

[4] The Galleria model does not really tell us anything new or useful about these strains. The model itself is an acute infection model so it is not surprising that PAO1 kills the larvae while biofilm forming host adapted clinical strains do not. I think the ideal model to investigate these clinical strains would have been a mouse model of chronic lung infection, one which allows persistence of infection, biofilm formation and interactions with both humoral and cellular immunity.

Response: We agree with the reviewer that a mouse model would have been adequate to determine virulence during chronic infection. However, our interest was to determine the safety of the phage therapy and the rationale for the virulence assays that we used was to show that the isolates after the phage treatment did not recover a more virulent (acute) phenotype, potentially due to integration of phage encoded virulence genes. In this respect our model is appropriate since the initial isolate (D-100) was already attenuated, due to long term adaptation to the host environment, as generally observed with isolates from CF-patients. On the other hand, we agree that we cannot show a further decrease in virulence. As stated above, the PAO1 reference strain is used to validate the Galleria assay rather than as a comparator for the patient isolates. Our own data (Luscher et al., Front Microbiol. 2020) and those from other groups seem to indicate that phage resistance, which we did not observe here, seems to select for attenuated virulence probably as a trade-off for phage resistance acquisition (Castledine et al., e-Life 2022).

Reviewer #4 (Remarks to the Author):

The authors report in details on a single case of phage use in a patient with an MDR pseudomonas infection. Although similar reports have been published before, it is refreshing to see such details in a case report.

My overall suggestions/comments:

1. The manuscript is extremely long for a case report, esp as a type of 'communication'. Some of the details may be more suited as a story, but may lack scientific value. I would

recommend:

(a) Reduction in number of, or details of previous studies on phage therapy: no need to review all the previous experience in a communication as the readers would be lost long before reading the actual report.

Response: We thank the reviewer for this comment. As suggested, we have significantly shortened the introduction section by removing detailed results of previous studies as well as the reference to ongoing phage therapy studies, which allowed reducing the introduction section from 500 to 323 words.

(b) Many details are not important: e.g. p127-132: specific dates are not important. If authors want to make a point on duration, they can change dates to number of days, which may be more useful in giving context to the 'story'. Or line 108-113: Can simply indicated phage therapy was initiated after consent was obtained from patient using phage vFB297... Words like 'last resort' makes the reading more entertaining but also distracting.

Response: This point is well taken. As stated already in the response to reviewer 2 (item 1), the patient history and treatment part has been reduced by 1/3 and is now equivalent to the length of the experimental work description. Together with the streamlining of the Introduction section, we believe that this significantly shortens the manuscript and improves its flow. We also removed mentioning of "last-resort".

(c) Authors should summarize the clinical course more succinctly, focusing not on providing as much detail as possible, but thinking about what is most important in their goal of the report.

Response: See our response to (a) and (b)

(d) Please comment on the phage safety: one of the main issues are quality control of phage and potential cytokines etc in the preparation

Response: This point has also been raised by reviewer 2 (item 3). We have added the following lines to the discussion section:

"The measured endotoxin concentrations were above the threshold levels indicated for IV application (5 EU/kg/h) and could explain the brief inflammatory response and fever episode after the first phage administration. However, the patient's chronic *P. aeruginosa* colonization and hence continuous exposure to *P. aeruginosa* LPS could explain that endotoxins present in the phage preparation were well-tolerated when administered by nebulization."

(e) Be careful in claiming 'quantitative' comparison in sputum samples: notorious that there are huge variability in content in different sputum samples (vs BAL, for instance) ???

Response: We acknowledge that clinical samples are difficult to compare "quantitatively". To minimize sample heterogeneity we did not compare different types of respiratory samples and only included "sputum" samples (excluding BAL for instance). In BAL the variability is indeed further increased by the dilution and varying relative fractions retrieved after NaCl instillation. However, there is in our opinion so far no better way to work with clinical samples as we tried using only sputum samples. To acknowledge the difficulty to perform quantitative comparisons, we have added a sentence in the discussion section limiting the interpretation:

“We also cannot exclude some variability in content of the sputum samples that might have affected quantification and sample comparison.”

2. There are quite a few minor points that are not too important but can make the reader keep pausing and affect the readability an otherwise enjoyable journey of a patient's story. A few examples:

(a) Line 87: no need to say 'congenital': this is understood

Response: We have deleted the corresponding sentence

(b) Line 88: situs inversus is not a 'complication' of Kartagener but really part of the syndrome

Response: We agree with the reviewer and have adapted and shortened the wording in the first paragraph of the results section and reformulated the first sentence:

“The patient was a 41-year-old male with Kartagener syndrome.”

(c) No need to report the doses of antibiotics

Response: As suggested by this reviewer, mentioning of the antibiotic dosages has been removed in the revised version of this manuscript.